# Classification and Identification of Spectral Pixels with Low Maritime Occupancy Using Unsupervised Machine Learning

**Dongmin Seo [1], Sangwoo Oh [1,*] and Daekyeom Lee [2]**

[1]   Maritime Safety and Environmental Research Division, Korea Research Institute of Ships and Ocean Engineering, Yuseong-daero 1312beon-gil, Yuseong-gu, Daejeon 34103, Korea; dseo@kriso.re.kr
[2]   SEASON Co., Ltd., Sejong City 20128, Korea; daek29@season.co.kr
*   Correspondence: swoh@kriso.re.kr; Tel.: +42-866-3615; Fax: +42-866-3624

**Abstract:** For marine accidents, prompt actions to minimize the casualties and loss of property are crucial. Remote sensing using satellites or aircrafts enables effective monitoring over a large area. Hyperspectral remote sensing allows the acquisition of high-resolution spectral information. This technology detects target objects by analyzing the spectrum for each pixel. We present a clustering method of seawater and floating objects by analyzing aerial hyperspectral images. For clustering, unsupervised learning algorithms of K-means, Gaussian Mixture, and DBSCAN are used. The detection performance of those algorithms is expressed as the precision, recall, and F1 Score. In addition, this study presents a color mapping method that analyzes the detected small object using cosine similarity. This technology can minimize future casualties and property loss by enabling rapid aircraft and maritime search, ocean monitoring, and preparations against marine accidents.

**Keywords:** hyperspectral imaging; maritime vessel detection; unsupervised machine learning; clustering algorithms; small object detection; color-mapping; aircraft remote sensing

## 1. Introduction

In recent years, the risk of marine accidents has risen because of an increasing volume of seaborne trade, other marine traffic, and marine-leisure activities. The importance of research on maritime vessel detection has garnered growing attention. To prepare for marine accidents and facilitate rapid response, studies on vessel detection using high-resolution optical images are underway [1–3]. Maritime vessel detection has evolved by building datasets with numerous images based on deep learning technology [4,5]. Prompt detection of ships is critical for efficient search and rescue operations and minimizing the loss of property. Therefore, remote sensing methods using satellites or aircrafts have been investigated for monitoring large areas efficiently [6].

Existing research on maritime vessel detection and tracking include studies on maritime object detection with synthetic aperture radar (SAR) and satellite images [7–10]. In addition, vessel detection has been studied by analyzing images acquired with an optical camera mounted on a satellite or by analyzing aerial images [6,11,12]. SAR is used to detect ships during the day and at night, despite severe weather with clouds, snow, rain, or fog. However, SAR is not suitable for the identification of a material based on its low spectral resolution.

There are studies on sea-land separation and detection of vessels for analysis of maritime vessels using images. Integrated research with sea-land-vessel separation has also been conducted, and on the analysis of maritime traffic and the development of new routes through the analysis of vessel trajectories [13–15]. However, since the above studies investigated image-based analysis techniques, their applications are restricted to the detection of objects with shapes that are larger than a specific size limit.

Hyperspectral remote sensing using hyperspectral imaging technology was introduced to improve the image-based shape analysis method. By analyzing spectral bands

with wavelength resolutions of the order of several nanometers, the technology allows the detection of objects that could not be otherwise detected in remote sensing based on RGB imagery. Thus, this technology has been utilized in the survey of land, minerals, and vegetation, and in chlorophyll analysis for water quality monitoring [16–24]. Recently, hyperspectral remote sensing has been employed and studied in various fields of oceanography. There have been studies on the analysis of shallow coastal topography below the water surface, and on the mapping of Arctic coastal cliffs and detection and classification of sea ice images [25–27]. In addition, the technology has been employed in the detection of hazardous noxious substances and autonomous surface vehicles with ocean color remote sensing [28,29]. In terms of hyperspectral remote sensing for vessel detection, studies have been conducted on vessel detection based on hyperspectral images and detection of floating objects, such as non-vessels and surfboards [30–33]. In addition, a hyperspectral neural network-based study for the detection of small objects has been conducted [34], wherein it was suggested that the hyperspectral dataset has superior detection accuracy in CNN-based result analysis compared with the RGB dataset for the detection of small objects that have an occupancy of approximately 1% compared to the background. This suggests that when using the learning technique, the more diverse the spectrum information obtained from a single pixel, the more helpful it is to analyze small objects.

Hyperspectral data consist of two spatial dimensions and one spectral dimension. To distinguish an object using the collected hyperspectral data, it is necessary to distinguish the spectrum of each pixel constituting the object. However, analysis using the entire spectrum is resource intensive due to the vast amount of hyperspectral data [35]. Therefore, dimensional reduction of hyperspectral images has been studied [27,31,36–38]. However, the general application of the method to different images that are subject to other conditions is difficult in practice because the dimensional reduction is highly sensitive to the acquisition environment of the hyperspectral images or impacts that may arise in the correction process. Unsupervised learning methods can serve as efficient alternatives to facilitate the spectral classification of pixels. Unsupervised learning is a field of machine learning that classifies big data into several categories to identify patterns without explicitly learning the correct answers. This technique is used in research where the amount of data is insufficient to conduct supervised learning or in research on object classification into clusters derived by comparing prominent features. Therefore, if the clustering technique of unsupervised learning is applied to the identification and classification of objects using hyperspectral data, the features of each pixel can be distinguished in detail using the whole spectrum of the pixel.

Herein, we propose a method of maritime object detection and identification in which the spectrum of each pixel in hyperspectral data is analyzed using a clustering algorithm to detect small target objects on seawater. The object on seawater is identified using the spectrum of ground-truth images for the target.

## 2. Materials and Methods

### 2.1. Analysis Procedure

Figure 1 shows an overview of the clustering process of hyperspectral data and the detection of objects on seawater (classification into seawater and objects on seawater) based on the clustering process. First, an arbitrary sea area is scanned using a hyperspectral camera mounted on an aircraft. Most of the area scanned by the aircraft is seawater, and objects floating in the sea are sparsely distributed. The spectrum information of each pixel is analyzed using a clustering algorithm and classified according to the identified characteristics of the spectrum. Small objects floating in the sea are detected by calculating the density of the classified clusters, and separating the seawater clusters with high densities from other clusters with low densities.

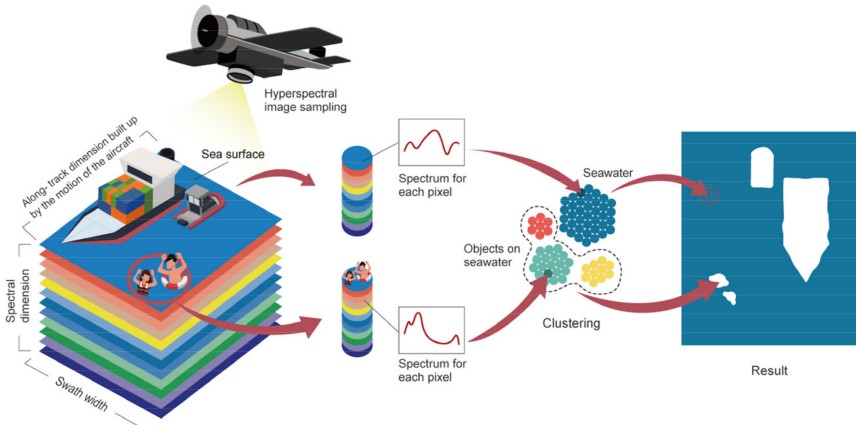

**Figure 1.** Schematic for the detection of small objects on seawater using aircraft. Objects on seawater are detected by scanning hyperspectral data using aircraft and clustering the scanned data using machine learning technology.

Figure 2 presents a flowchart for the detection and identification of small objects on seawater using hyperspectral data. Aerial hyperspectral data requires preprocessing because distortion occurs due to electromagnetic waves, geometrical errors, and atmospheric conditions. The preprocessing of hyperspectral data incudes radiometric correction, geometric correction, and atmospheric correction [38]. Herein, hyperspectral data with radiometric correction and atmospheric correction were used. Radiometric correction is processed with the SHIPS software (SPECTIF, Sydney, Australia). Atmospheric correction is performed using ATCOR 4 based on MODTRAN 5 (MODerate resolution atmospheric TRANsmission 5). After the application of the correction processes, the pixels of the hyperspectral data are clustered with machine learning algorithms, such as K-means, Gaussian Mixture, and density-based spatial clustering (DBSCAN). When the pixels are clustered using the respective clustering models, the clusters of pixels that can be inferred as seawater and objects on seawater are displayed as binary images. Through this process, the detection of objects on seawater is performed. This study used two types of coloring methods: the hyperspectral RGB coloring method and the use of the spectrum of ground-truth images as a reference and mapping with the most similar color. For color mapping with the most similar color, the spectral angle mapper (SAM) method that uses cosine similarity was applied.

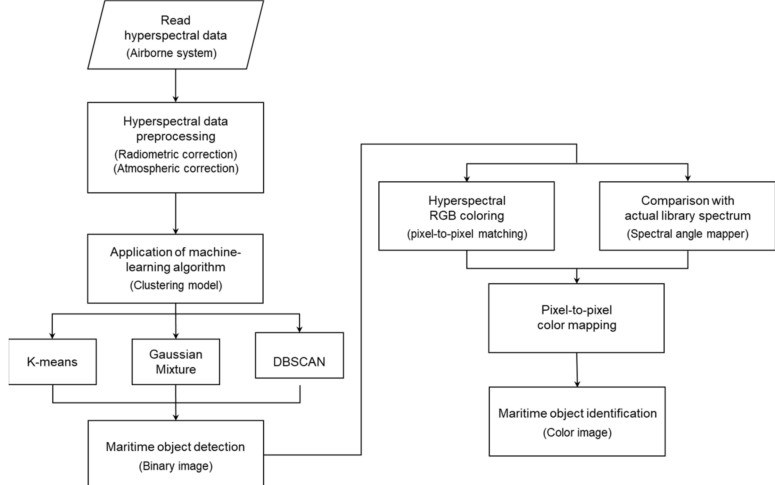

**Figure 2.** Flowchart of the data analysis process for the detection and identification of small objects on seawater. For the detection of objects on seawater, hyperspectral data are analyzed with a clustering algorithm. For the identification of objects on seawater, color mapping is performed on the detected objects. The methods of color mapping include RGB color matching and reference color matching.

*2.2. Clustering Techniques*

In this study, the clustering algorithms used to detect marine floats are K-means, Gaussian mixture, and DBSCAN. The K-means algorithm groups the given data into K clusters and operates in a way that minimizes the variance of the difference between each cluster and distance [39,40].

The K-means algorithm proceeds in the following steps [41]:

(1) Randomly place K centroids.
(2) Allocate each data point to the nearest centroid (clustering).
(3) Update the centroid of the cluster based on the data designated as a cluster.
(4) When the centroid is updated, the new centroid is updated with the average value of the assigned data.
(5) Repeat steps 2 and 3 until the new centroid is not updated.

In this study, the K-means++ algorithm is applied. K-means++ is an improved method to address the issue of the K-means algorithm randomly selecting the first centroid setting [42]. K-means++ randomly selects one of the data points and designates it as the centroid. Then, for the remaining data points, the distance to the first centroid is calculated. The centroid of the next sequence is designated as a data point arranged as far as possible from the first designated centroid. Finally, iterations are run until K centroids have been selected and then the same step as K-means are followed. Besides this step, the procedure is the same as that of other K-means algorithms. For the options used in the experiment, the default values were used except for the cluster values.

The Gaussian Mixture Model (GMM) can be described as a weighted sum of N Gaussian components [43]. The GMM is a model that assumes that the distribution of the data constituting the whole is generated from the distribution of N subordinate data following a normal distribution. There are two types of parameters in GMM, and data are classified through parameter estimation. The first type is a weight value that indicates where it belongs among the N normal distributions, and the second type comprises the parameters (mean, variance) of each normal distribution. These can be seen as the parameters of GMM, which are estimated from the training data using the EM algorithm or a well-trained maximum a posteriori (MAP) estimation [44]. After the parameter estimation is completed, it is possible to probabilistically determine the distribution from which the observed data was derived. The observed data is classified as a normal distribution with the highest probability. In the GMM applied in this study, Scikit-learn's Gaussian mixture algorithm was used, and default values were applied to the option values, excluding the component values.

The DBSCAN algorithm is based on density reachability and density-connectivity [45,46]. The core idea of DBSCAN is that points are assigned to the same cluster if they are density-reachable from each other. The DBSCAN algorithm consists of key parameters, such as EPS, MinPts, core point, and border point. EPS denotes the distance from the data point. MinPts is the number of data points to be recognized as a cluster in the EPS. The core point is a point when data is included with at least MinPts number within the EPS distance. A border point is a point that is included in the cluster but does not become a core point. The DBSCAN algorithm follows the following steps [47].

When EPS and MinPts conditions are given as input parameters:

(1) Select an arbitrary point that satisfies the condition of core points as an initial value (seed) in the spatial dataset.
(2) Separate the core points and border points by extracting density (base)-reachable points from the initial value, and classify points that do not belong to these as noises.
(3) Connect the core points located around a circle of radius EPS.
(4) The connected core points are defined as one cluster.
(5) All border points are allocated to one cluster (if the border point spans multiple clusters, it is allocated to the first cluster in the iterative process).
(6) The algorithm repeats the above steps and terminates when no more points can be allocated to clusters.

In this study, Scikit-learn's DBSCAN was used, and default values were applied to all optional values except the EPS value and algorithm. For the algorithm option, the brute parameter was applied considering the response speed.

### 2.3. Color Mapping Techniques

Spectral angle mapper (SAM) is a technique used to find the similarity of the spectrum using the cosine similarity. Using the dot product between the reference spectrum vector and the image spectrum vector, the degree of similarity is determined by the angle formed by the two vectors. The value of SAM is expressed in radians, where the minor angle $\alpha$ represents the major similarities between curves [48]. The mathematical formula of SAM is (1):

$$\alpha = \cos^{-1} \frac{\sum AB}{\sqrt{\sum (A)^2 \sum (B)^2}} \tag{1}$$

$\alpha$: The angle between $A$ and $B$.
$A$: Image spectrum value.
$B$: Reference spectrum value.

Assuming that the value of the image and reference spectra is $A$ and $B$, respectively, the angle $\alpha$ formed between the two spectral values is calculated. If the cosine similarity is $-1$, the direction of the vectors is opposite. The cosine similarity of 0 indicates that the vectors are orthogonal. The cosine similarity of 1 means that the vectors have the same direction. However, since the feature vectors have positive values, the cosine similarity shows values between 0 and 1. Two spectra are more similar when the result is closer to 1.

### 2.4. Airborne Hyperspectral Data

The images used for the experiment were taken in the waters near the Saemangeum Embankment located in Gunsan, South Korea (Figure 3). Stable low-altitude images were captured using a medium-sized single-engine aircraft (Cessna Grand Caravan 208B), maintaining a cruising speed of no more than 260 km/h, at an altitude of approximately 1 km. A digital mapping camera (DMC) and a hyperspectral sensor, installed on the aircraft, were positioned to take pictures from the bottom of the aircraft. The DMC (digital mapping camera, Intergraph's Z/I Imaging, Aalen, Germany) is a high-resolution digital camera for aerial surveying and is equipped with a 3072 × 2048 pixel RGB sensor. The hyperspectral sensor (AisaFENIX sensor, Specim, Oulu, Finland) can scan wavelengths in the 400 to 990 nm range with 127 bands. The spatial resolutions of the DMC and hyperspectral sensor are approximately 0.10 and 0.70 m, respectively, at an altitude of approximately 1 km. The two instruments can simultaneously capture images of a certain area during aerial operation. Therefore, a high-resolution DMC was used for clear identification of the target object captured by the hyperspectral sensor. Figure 3 shows the arrangement of 7 vessels photographed by DMC; the position of the upper left vessel shown in the figure is approximate, latitude: 35°54′43.24″N, longitude: 126°31′6.66″E.

Figure 4 presents hyperspectral RGB images of the area marked in Figure 3. The hyperspectral RGB image was obtained by extracting and combining the red wavelength of 640 nm, green wavelength of 547 nm, and blue wavelength of 470 nm out of a total of 127 bands. Seven vessels numbered #1 to #7 are shown, and the zoomed-in images of vessels taken with DMC are matched and displayed. The number corresponding to each image is used consistently in all the figures. Vessel #1 is a recreational boat with a red deck and white cabin and hull. Vessel #2 is a typical fishing boat that is mostly blue except for the white cabin. Vessel #3 is a 3-ton fishing vessel without a cabin and is mostly blue. Vessel #4 is an emergency rescue boat made of black rubber with a red cabin. Vessel #5 has a gray deck and a white-and-purple cabin. It is used as a recreational fishing boat. Vessel #6 has a similar shape to vessel #2, but a brown wooden deck is placed on the bow side. Vessel #7 is mostly red and is used as a recreational fishing boat. Except for Vessel #2, the other 6 vessels are floating in the sea. Vessel #2 generates white waves depending on the

direction of its motion. Small maritime objects floating in the sea are connected by ropes to vessels #6 and #7. For Vessel #6, a 6-seater life-raft, a 12-seater buoyant apparatus, and 2 mannequins in the standing positions are placed overboard. For Vessel #7, 3 lifebuoys and 2 mannequins in the supine position are placed overboard. In total, there are 16 maritime objects—7 vessels and 9 small maritime objects placed overboard from 2 vessels—that are used for evaluating the detection performance.

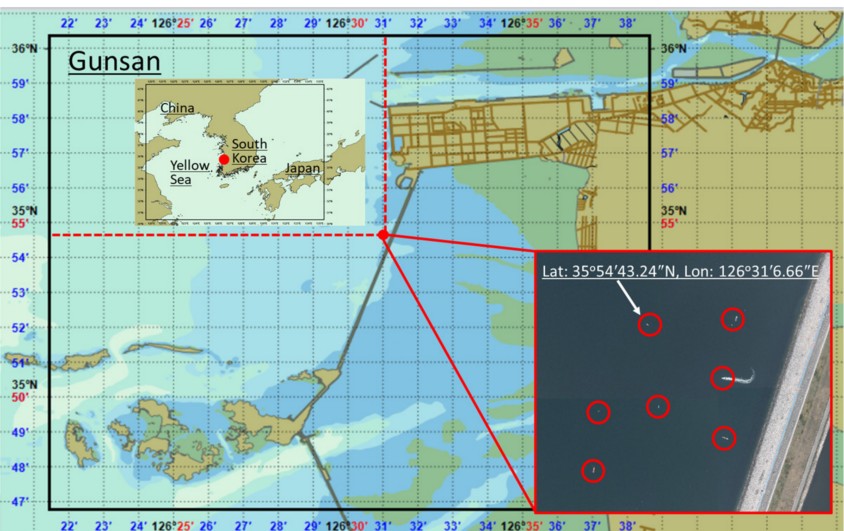

**Figure 3.** Location of the offshore field experiment for hyperspectral image acquisition using the airborne system equipped with high-resolution optical and hyperspectral cameras. Aerial images were taken at an altitude of 1 km in the waters near the Saemangeum Embankment in Gunsan City, South Korea.

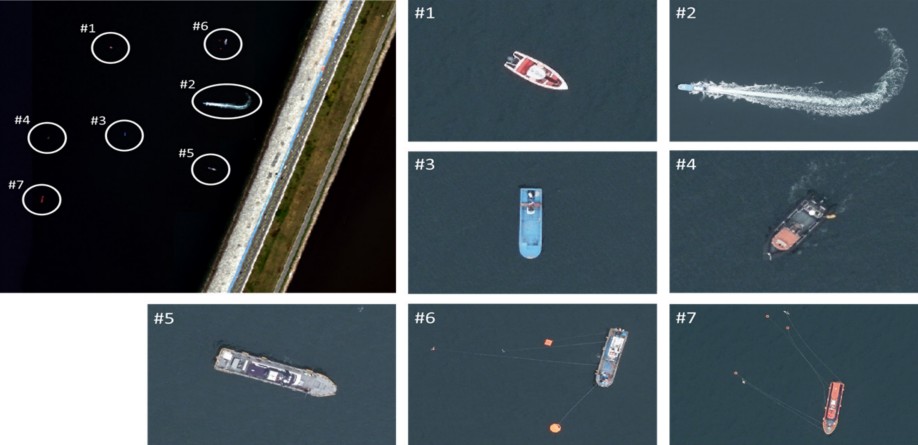

**Figure 4.** DMC images of small vessels and objects on seawater used in the experiment. Seven small vessels, #1 to #7, were used for the experiment; overboard from vessels #6 and #7, life-saving appliances, and a human mannequin are floated. The numbers (#1 to #7) corresponding to each target object are used consistently in the figures that follow.

*2.5. Field Spectrum Data*

The field spectroradiometer (ASD FieldSpec 4 Wide-Res, Malvern Panalytical, Malvern, England) used herein has an observation range of 350 to 2500 nm and a sampling performance of 1.4 nm below 1000 nm, and 1.1 nm otherwise. This instrument measures the reflectance of a target object using a handheld device with a pistol grip design combined with a fiber optic cable. For measurements, the target object is pointed at with the pistol at an angle of 30 degrees above the ground. A total of 18 spectral references, including

seawater, were obtained using the field spectroradiometer (see the Supplementary Materials Figures S1 and S2). The observation timestamp was 11 A.M. on 23 May 2021, and the weather conditions were clear. These data were used for the identification process of this study to map the pixels detected by the clustering algorithm into objects. Clustered pixels were analyzed by SAM and assigned a color with one reference spectrum. However, only those in 400 to 990 nm, which is the spectrum range of the aerial hyperspectral sensor, were extracted from the reference spectrum and used for the identification of maritime objects.

## 3. Results and Discussion

Figure 5 shows the results of the clustering algorithm models in detecting the 5 types of vessels in the sea. For intuitive analysis of the detection accuracy of the maritime objects removed from seawater, the RGB images and ultra-high-resolution DMC images of the hyperspectral data were added. For K-means and Gaussian Mixture, K was set to 2 or 3. From the results of the detection analysis, a K of 2 was found to be optimal for the detection of maritime objects. In the case of DBSCAN, the epsilon (EPS) needs to be set according to the range of data to be analyzed. The detection performance of DBSCAN was determined based on the EPS. To derive the optimal EPS value of DBSCAN, EPS values were selected while increasing it from 0.015 to 0.045 in steps of 0.01. The optimal EPS of DBSCAN for the detection of maritime objects in images of a mostly seawater region is 0.025.

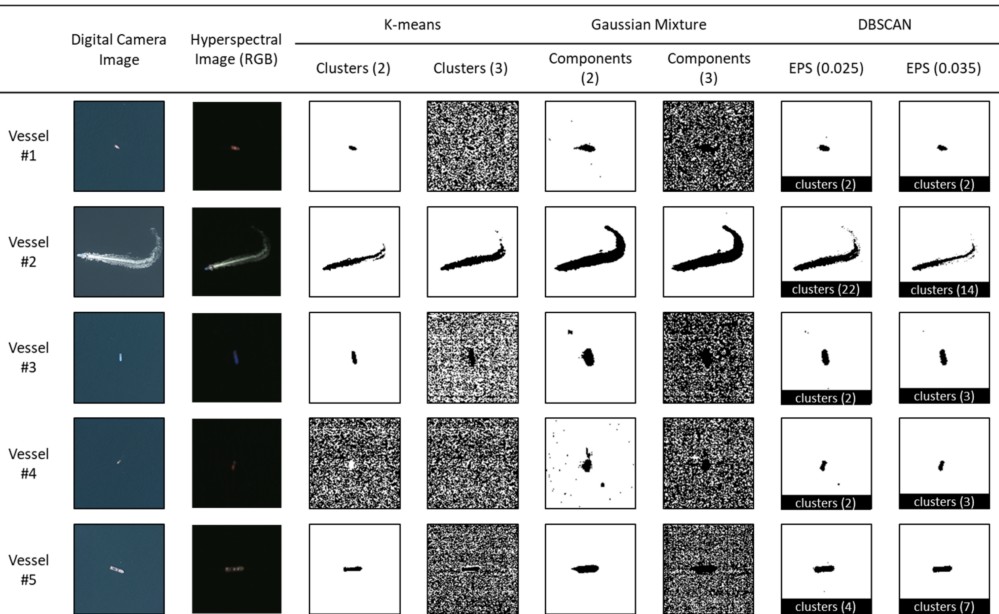

**Figure 5.** Small vessel detection performance analysis using clustering algorithms. The detection performance for five target vessels in the sea was analyzed using K-means, Gaussian Mixture, and DBSCAN as representative clustering algorithms in machine learning. Target detection refers to the detection of pixels of maritime objects excluding seawater clusters. Ultra-high-resolution DMC images and hyperspectral RGB images were added to facilitate intuitive viewing and checking of the detection results for each target.

Analysis of the detection results of the three clustering algorithms showed that the performance of Gaussian Mixture in detecting maritime vessels is inferior to that of other algorithms. Though it appears to perform reasonably at K = 2, the result shows more noise than other algorithms, and the boundary between the target object and seawater is unclear. Conversely, K-means shows superior vessel detection performance compared to Gaussian Mixture. However, in the detection result of vessel #4, a clear differentiation between seawater and maritime objects was not possible. In contrast, DBSCAN succeeded in maritime object detection by analyzing clusters of the seawater and the objects on the seawater with EPS at 0.025 and 0.035. In particular, in the results detecting targets #3, #4,

and #5, the outcome with EPS at 0.025 shows a better performance in differentiating lower-density pixels, such as buoys and wave breakers, than the result with EPS at 0.035. Thus, DBSCAN's detection results contain less noise than Gaussian Mixture and its clustering performance is better than K-means for the detection of maritime objects in pixel units.

Figure 6 shows in detail the performance of each model in clustering and classifying seawater and maritime objects by classifying the spectrum for the #4 target and composing the image with the classified spectrum. Vessel #4 was chosen because it had the lowest detection performance. Spectral clusters with index 0 are colored blue and those with index 1 are colored orange. The x-axis of the graph represents 127 spectral bands, and the y-axis represents reflectance. Dark solid lines in blue and orange indicate the average of each clustered spectrum, and an error bar indicates the standard deviation between pixels. Here, it is reasonable to consider that an index with a large number of pixels represents the seawater cluster, and an index with a small number of pixels represents the maritime object cluster. In the image composed of the indices, the pixels corresponding to the indices are expressed in white.

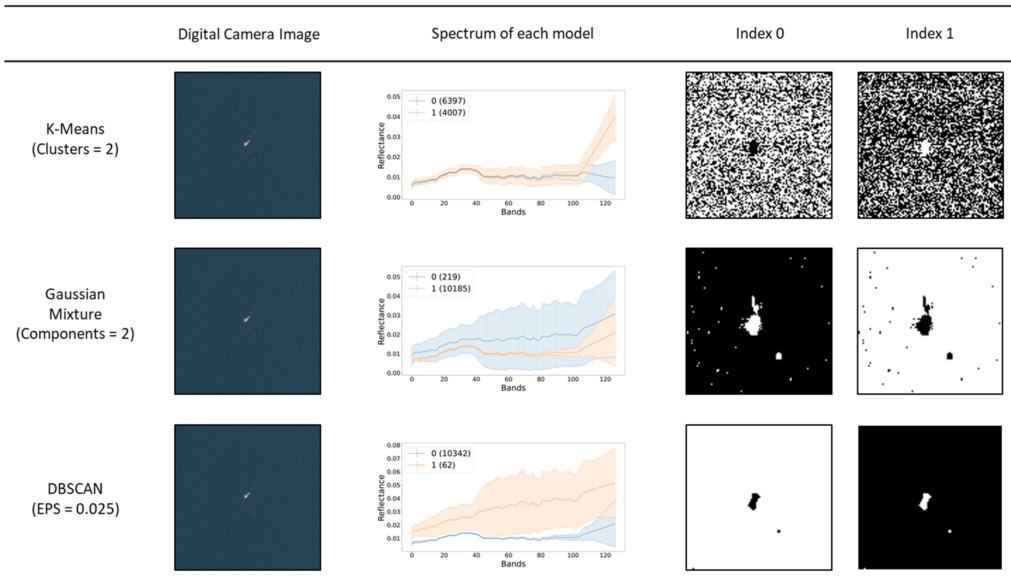

**Figure 6.** Spectral clustering results of the clustering algorithms. Using target #4 from Figure 5, the spectral clustering results of each clustering model are presented in graphs and images. The area corresponding to the index is indicated in white.

First, from the spectrum of K-means, 6397 pixels are classified into index 0 and 4007 pixels into index 1. The spectral average lines of the 2 clusters are almost identical except for the band in the range of 110–120. Therefore, if each index is displayed as an image, the seawater part is not clearly classified, and the seawater area is divided into black and white. This is the result of forcibly classifying the spectrum that should be judged to be the same cluster by setting the K to 2. However, if the K is changed to other values to optimize the detection in this image, the performance would decrease for the detection of other maritime objects. Examining the spectrum of the Gaussian mixture, 219 pixels are classified into index 0 and 10,185 pixels into index 1. The spectral average lines of the 2 clusters show a difference of more than 0.005. However, looking into the standard deviation between pixels constituting each band, the spectra of the two clusters still overlap in the entire area. This result indicates that even in the Gaussian mixture, the pixels of seawater and maritime objects cannot be completely differentiated. Examining the image composed of each index, a significant amount of noise is generated in some seawater areas and at the boundary of maritime objects. Conversely, in the spectrum of DBSCAN, the spectral average lines of the two clusters are clearly distinguished. The number of pixels in index 0 is 10,342, and the number of pixels in index 1 is 62.

These results imply that, in DBSCAN, the performance of Euclidean distance analysis that distinguishes the spectrum between seawater and maritime objects is superior to that of K-means and Gaussian mixture. However, in the spectrum classification graph of DBSCAN, overlapping parts are identified around band 120. The reflectance of this part is approximately 0.025–0.035, and there is a possibility that a pixel detection error may occur depending on the EPS. The number of pixels of the maritime object is only 62 compared to the number of pixels of seawater. Therefore, even if noise increases slightly, it can be reconfirmed that setting EPS = 0.025 is the suitable parameter for analysis of those images. The number of clusters changes according to the EPS and is equal to or greater than the optimal cluster range for the K-means and Gaussian Mixture algorithms.

Figure 7 shows the performance of the clustering algorithms in detecting marine objects smaller than the size of the vessels by detecting life-saving systems and mannequins mimicking a man overboard. In vessel #6, a 6-seater life-raft and a 12-seater buoyant apparatus were placed, and 2 mannequins in the standing position were placed overboard. In the red vessel #7, 3 lifebuoys and 2 mannequins in the supine position were placed overboard. The material was not considered a factor in this study because the mannequin mimicking the man overboard was used as a dummy for the detection of small objects according to the size and shape. The optimal value derived from Figure 5 was used as the K of the clustering algorithm used for the detection.

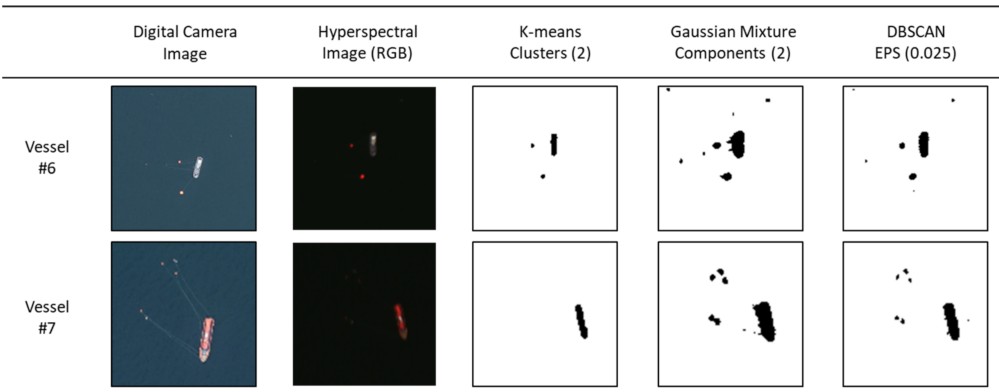

**Figure 7.** Detection performance analysis of small objects on seawater using a clustering algorithm. The detection performance of two target vessels in the sea and small objects overboard was analyzed using clustering algorithms. In the target of #6, the vessel, buoyant apparatus, life-raft, and dummies in the standing position, and in the target of #7, the vessel, lifebuoys, and dummies in the supine position, are connected with ropes and placed overboard with a certain distance between them. Ultra-high-resolution DMC images and hyperspectral RGB images were added to facilitate intuitive viewing and checking of the detection results for each target.

Examining the detection results using the clustering algorithm, K-means detected the vessel, life-raft, and buoyant apparatus in the target vessel #6. However, they did not detect the mannequins in the standing position. Conversely, in the case of vessel #7, the vessel was detected, but neither the buoy ring nor the mannequin in the supine position was detected. These results indicate that the clustering performance of K-means in the detection is significantly reduced for an object under a certain size. In the case of the Gaussian mixture, the vessels, life-saving systems, and mannequins used in #6 and #7 were all detected. However, when compared to the DMC image, the detection result has noise, wherein the boundary of the detected object appears larger than the actual object or spreads out. In the presence of this type of noise, identification of the shape of the vessel is more complicated. In addition, as visible from the detection results of the lifebuoy and the mannequin in the supine position not wearing a life-saving appliance in target #7, the resolution is reduced, and the object is recognized as a single object. DBSCAN detected all

maritime objects used in the experiment except for the mannequin in the standing position wearing a blue life jacket in #6.

In summary, K-means shows the poorest performance in detecting small maritime objects. Although the Gaussian Mixture detected smaller marine objects compared to DBSCAN, its clustering performance was poor; therefore, the resulting image appeared noisy. Table 1 shows the best performance at K = 2 for K-means, $n$ = 2 for Gaussian Mixture, and EPS = 0.025 for DBSCAN. Among those algorithms, DBSCAN has the highest F1 Score of 0.9649 at EPS 0.025.

**Table 1.** Statistical analysis of the clustering algorithms. Algorithm performance is expressed as precision, recall, and F1 Score.

| | K-Means | | | | Gaussian Mixture | | | | DBSCAN | | | |
|---|---|---|---|---|---|---|---|---|---|---|---|---|
| | K = 2 | K =3 | K = 4 | K = 5 | $n$ = 2 | $n$ = 3 | $n$ = 4 | $n$ = 5 | EPS 0.015 | EPS 0.025 | EPS 0.035 | EPS 0.045 |
| Precision | 0.2886 | 0.0703 | 0.0395 | 0.0391 | 0.1562 | 0.0547 | 0.0392 | 0.0354 | 0.5348 | 0.9322 | 0.9480 | 0.9599 |
| Recall | 0.6852 | 0.9155 | 0.9067 | 0.9433 | 0.9739 | 0.9996 | 1.0000 | 1.0000 | 1.0000 | 1.0000 | 0.6587 | 0.4733 |
| F1 Score | 0.4061 | 0.1306 | 0.0757 | 0.0751 | 0.2692 | 0.1307 | 0.0754 | 0.0684 | 0.6969 | 0.9649 | 0.7773 | 0.6340 |

Figure 8 shows the result of the color mapping using spectral angle mapper (SAM) for the pixels of the maritime object detected using DBSCAN. First, images acquired from ultra-high-resolution DMC are placed at the top of the figure to help compare the results of each analysis. A hyperspectral RGB map, in which the spectrum of each pixel detected by DBSCAN was mapped to the hyperspectral RGB pixel spectrum, was placed in the middle. The DBSCAN reference map set at the bottom is the result of replacing the spectrum of each pixel obtained by detection with DBSCAN with the color of the reference spectrum and with a high cosine similarity. A list detailing the reference color settings is presented in Figures S1 and S2.

As a result of reference mapping, in target #1, the deck of the vessel was seen in brown, blue, and gray. The color composition of the deck correctly showed the color composition of the actual vessel, but the proportion of brown was higher than that of blue. The white part of the cabin was also clearly presented according to the reference color in gray. The life-raft and buoyant apparatus were indicated in orange and red, respectively, and the dummies in the standing position were shown in brown. In the case of #2, the colors of the white part and the red part of the vessel were clearly shown. However, the red in the bow was not expressed. In #3, the blue deck of the vessel was clearly identified, and the white cabin was also mapped according to the reference. The center of the wave at the stern of the vessel was classified as seawater. This indicates that breaking waves can also be classified through spectrum analysis. However, since most waves were classified as gray, additional research is needed for a more refined analysis. In #4, the characteristics of the vessel colors composed of white, brown, and blue were presented similarly, and in #5, a vessel characterized by a blue deck, the colors of the blue deck and brown stern were expressed similarly. In the case of the rubber boat, target #6, the mapping results showed the colors red, brown, and gray. Among these colors, the expression of red corresponding to a bow is accurate. However, since there is no reference spectrum corresponding to black, it seems to be expressed in brown with the highest similarity of 0.99 and gray (0.96) with the second-highest degree of similarity compared to black. In the case of #7, the color of the vessel was similar to that of the actual vessel, but the boundary of the vessel was expressed in brown. This might be caused by the seawater spectrum at the interface affecting the red spectrum of the vessel, increasing the similarity with the brown spectrum. The life-buoy and mannequins were expressed in brown and could not be distinguished by color. It is presumed that these small objects also showed high similarity to the brown reference spectrum due to the interference of the seawater spectrum.

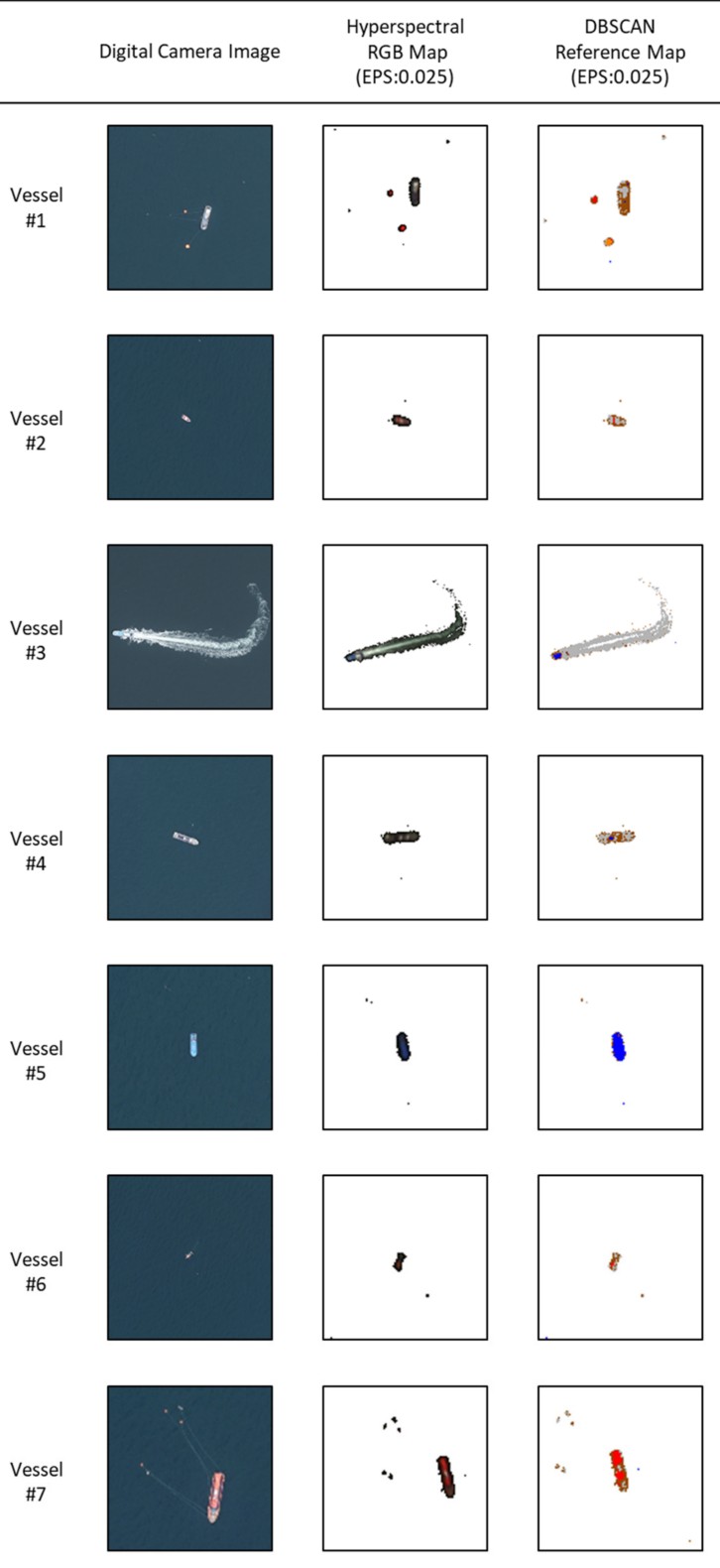

**Figure 8.** Coloring method of the detected pixels using cosine similarity. The pixels of the maritime object detected by DBSCAN are compared with the reference spectrum and replaced with color in high similarity (DBSCAN Reference map). Ultra-high-resolution DMC images and hyperspectral RGB map images that applied hyperspectral RGB color mapping to the DBSCAN detection results were added to facilitate intuitive viewing and checking of the color mapping results for each of the targets.

In summary, compared to the hyperspectral RGB map acquired using a hyperspectral camera, the color mapping using the reference spectrum (DBSCAN reference map) shows improved results in distinguishing the colors of objects. In addition, each pixel can be mapped to the objects for classification, and the method allows the identification of some objects. However, the noise generated at the detection boundary and errors owing to the reconstruction of reflected waves due to low-resolution images are some persisting challenges.

## 4. Conclusions

In this study, we presented a maritime object detection and identification method in which a maritime object is detected in a hyperspectral image using a clustering method, and the detected maritime object is identified through reference mapping of the detected pixels. Three types of clustering techniques, K-means, Gaussian Mixture, and DBSCAN, were used for seven small vessels, and the detection performance of each method was comparatively analyzed. In addition, life-saving systems and man-overboard mannequins were used to examine the detection performance for small objects on seawater. The detection performance of the algorithms was compared in terms of precision, recall, and F1 Score. The best performance was achieved at EPS = 0.025 of the DBSCAN algorithm. In addition, color mapping was performed by comparing the cosine similarity between the spectrum of each pixel detected as a maritime object and the ground-truth reference spectrum. Thus, mapping of the detected pixels could be used to identify objects. However, since the ground-truth reference spectrum data was highly limited in availability, there was a limit to identifying the maritime objects. If the quantity and quality of comparable reference data can be improved, it is expected that more accurate object identification may be possible. Future development of the technology is expected to be utilized in key maritime activities, such as search and rescue, monitoring, and preparation against marine accidents.

**Supplementary Materials:** The following supporting information can be downloaded at: https://www.mdpi.com/article/10.3390/rs14081828/s1, Figure S1: A list detailing the reference color settings, Figure S2: Spectral data of reference color.

**Author Contributions:** Conceptualization, D.S. and S.O.; methodology, D.S.; software, D.L.; validation, D.S., D.L. and S.O.; formal analysis, D.S.; investigation, D.L.; resources, S.O.; data curation, S.O.; writing—original draft preparation, D.S.; writing—review and editing, S.O. and D.L.; visualization, D.S.; supervision, S.O.; project administration, S.O.; funding acquisition, S.O. All authors have read and agreed to the published version of the manuscript.

**Funding:** This research was supported by a grant from Endowment Project of "Development of hyperspectral image analysis technology for rapid detection and identification of marine accidents based on machine learning approaches" funded by Korea Research Institute of Ships and Ocean engineering (PES4460).

**Data Availability Statement:** Data underlying the results presented in this paper are not publicly available at this time but may be obtained from the authors upon reasonable request.

**Acknowledgments:** The authors thank Yeonghun Chae for their invaluable technical support.

**Conflicts of Interest:** The authors declare that they have no known competing financial interest or personal relationship that could have appeared to influence the work reported in this paper. The funders had no role in the design of the study; in the collection, analyses, or interpretation of data; in the writing of the manuscript, or in the decision to publish the results.

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
