# Peer review of "Classification and Identification of Spectral Pixels with Low Maritime Occupancy Using Unsupervised Machine Learning"

_remotesensing, doi:10.3390/rs14081828_

Round 1
Reviewer 1 Report
line 141: did you scale the data when applying the distance between data points? If so, how did you scale them. Thanks
Reviewer 2 Report
The authors use hyperspectral imagery of vessels and other objects to evaluate the potential of multiple clustering methods in unsupervised object detection. They compare k-means, Gaussian mixture models and the DBSCAN algorithm based on seven sample scenes.
The manuscript is overall well-structured and written concisely. However, the methodological description is very vague and lacks important information to support understanding of the workflow. Furthermore, the exact motivation and reasoning of the authors is not clear and some parts of the manuscript need rearranging.
Below, I list my main comments followed by minor points that mostly refer to aspects of style or language.
Line numbers are indicated at the beginning of each comment. Parts taken from the manuscript are italicized.
- 38ff: here you are jumping from discussing vessel detection to the survival time of people on open sea to the need of remote sensing methods for monitoring. Please improve the paragraph to avoid such logical jumps.
- 51ff: at this point you start discussing hyperspectral imaging technology but it is not clear why you are using it. So far, you have not explained the advantages of hyperspectral imagery over regular RGB or multispectral imagery for this particular application. Judging from the example images you give later in the manuscript, I would suspect that multispectral imagery would work just as well in detecting those vessels (of course depending on the available wavelength bands). Please provide a better explanation on why hyperspectral imagery is needed here.
- Sections 2 & 3: there are several main points to be addressed here:
- Section 2.2: this section is crucial and should be improved. The three methods are described vaguely and it reads more like the function description of Scikit-Learn (which is also used as a reference). Please improve the explanation and refer to relevant literature.
- You have not specified the way you selected the hyperparameters (number of clusters, epsilon). I assume it was by trial and error?
- I believe it would be valuable to evaluate cluster numbers k > 3 since the “background” (i.e. sea surface) might be heterogeneous due to waves and sea spray. Especially for k-means, it might be interesting to see performance with 4 or more clusters.
- In connection to (b): DBSCAN is fundamentally different from the other two algorithms in that it determines the number of clusters automatically. This is probably also a reason for the higher robustness observed here. It would be important to know how many clusters DBSCAN distinguishes in your dataset and then compare it to k-means and GMM with the same number of clusters set beforehand.
- Aside from the qualitative analysis you provided, there needs to be a quantitative analysis based on common classification metrics to better understand the differences in performance between the models and in what context they fail.
- 111ff: this section lacks an explanation of the correction and pre-processing steps undertaken. You mention atmospheric and radiometric correction but do not specify how you corrected the data.
- 164ff: this part has a different context than the rest of the section (SAM for visualization vs clustering techniques) and should therefore be separated.
- 169ff: you first describe values between -1 and +1 and then in the next sentence say resulting values are between 0 and 1. Please clarify.
- 235: do you mean you mapped the clustered pixels to classes by matching them to your spectral library through SAM? Please expand this part and clarify the procedure.
- 239ff: this part contains some duplicate information that has already been provided elsewhere. The parts describing the algorithms belong to the methods section, not results.
- 252f: what is meant by “changing the EPS from the first to the third decimal place”?
- 322f, 365ff & 435ff: these are very broad statements that go too far. This study is based on a small number of example images with a very specific ground truth setup. The results do not justify a general judgment of the algorithms or their use in maritime observations.
- 335ff: this paragraph belongs to the data description and is too detailed.
- 354: what do you mean by “resolution of K-means”? You refer to resolution multiple times in the manuscript (e.g. ll. 367 and 426) but it is never clarified what type of resolution you mean. Here, it sounds as if the clustering performance is referred to as resolution.
- 387ff: as far as I can tell, this part is just about the way you visualized the vessels in this specific context. I think it is not necessary to go into so much detail on how colors were selected or assigned. It would be different if it was your intention to also introduce a way to visualize results in real-world applications. In that case, make it part of your study objectives.
Figures and tables
- Figure 2: I think this figure was cut off at the top. Moreover, please list the performed preprocessing steps.
- Figure 8: the images are very small and details are difficult to see. This is the case for previous figures as well but here it is particularly obvious since you are describing the coloring which is sometimes difficult to distinguish in those examples.
Language and style
- 16: “…vast amount of spectral data…”
- 32: I think the comma is superfluous.
- 33: “…growing attention. To prepare…”
- 34: “…factilitate rapid response studies…”
- 71: “…due to the vast amount…”
- 378: “…targets.”
Miscellaneous
- 43: what do you mean by “image frames”?
- 47: what is meant by “candidate’s vessels”?
- 98ff: this information is basically recursive. Hyperspectral data is defined by its high spectral resolution. To say hyperspectral data contains spectral information for each pixel because it is multidimensional is not really informative.
- 112f: I think it is not necessary to specifically explain that the data is loaded onto a PC.
- 113ff: this sentence is difficult to understand. Please rephrase it.
- 123: I think this sentence can be removed.
- 312f: what do you mean by this sentence?
- 388: what do you mean by “18 spectrum data”?
- 401: what do you mean by “interest” here?
- 403: I think what you mean is “breaking waves”.
Reviewer 3 Report
General Comments:
I reviewed the manuscript entitled "Classification and identification of spectral pixels with low maritime occupancy using clustering algorithms in machine learning ". In this paper, Seo et al. have used the unsupervised machine learning-based method to identify the objects on the seawater. Further, this study utilized the spectroradiometer data to verify the algorithm results. This is a promising study that would help in advancing and detecting the maritime object remotely. However, this study is lacking with statistical analysis of the presented methods as well as the novelty of the present research. Also, authors should provide more details of ground-truth reference spectrum data as well as atmospheric correction approach was applied on it.
Here I am giving brief remarks on the current form of a paper.
Please see the comments and suggestions included in the following paragraph.
Title
I suggest replacing the study topic with “Classification and identification of spectral pixels with low maritime occupancy using unsupervised machine learning”.
Abstract
The abstract is not clear and lines 19-25 are the main parts of it. Therefore, I suggest authors revise the abstract for a better understanding and remove the lines which are not important to mention in the abstract.
Introduction
The introduction part is very well presented. The authors have carefully cited the work of the most recent research done in this area. However, I authors haven’t mentioned anything about the advantage of hyperspectral remote sensing over SAR remote sensing. SAR remote sensing is very popular for detecting an object on the ocean surface. Also authors should avoid repetition of sentences, thus, please revise the last paragraph of the introduction part and keep only the objective of this study.
Materials and Methods
This section needs to be improved because I couldn’t get the full details of the data and its processing methods.
A few suggestions are given as follows:
Subsection 2.3: The resolution of figure 3 is extremely low including the font size. Please use the better plot for the representation purpose.
Subsection 2.4: Please mention the measurements timing and sky conditions.
Please elaborate on the steps of converting the reflectance to radiance values. Authors should also plot and present the measured radiance spectra at all 19 stations. Further, emphasize the wavebands were used for the identification of the maritime objects.
Lines 232-234: This sentence is not clear to me “If the atmospheric condition is measured and the atmospheric correction is performed at the same position as the measured object, reflectance can be obtained”. Please rephrase it for clarity.
Results and discussion
Clustering algorithms are already mentioned in line 138. Kindly remove the details which are not relevant to the results, here, authors should present results and discuss them accordingly. I don’t find it worth mentioning the lines 240-246 “The clustering algorithms used (K-means, Gaussian Mixture, and DBSCAN) were provided by Scikit-learn which is a free software Python machine learning library that provides ten representative clustering algorithms [45]. The three algorithms were chosen due to their suitability for the analysis of Euclidean distances between spectra of pixels acquired at the maritime domain and for fast computation speed. The optional parameters of each clustering module were set to obtain expected values with excellent detection performance”.
Overall, this section is very well written. Results are very well presented and supported by neat and clean plots with the statistical matrices. The significance of the results is discussed very well in this manuscript. However, statistical analysis of the clustering algorithms is missing. Authors could rank the performance of the algorithms in descending or ascending order.
Conclusions
This section is very well written. The findings of the present study are articulated in a logical and organized manner.
Round 2
Reviewer 2 Report
The authors have improved the manuscript and addressed most of the comments.
Below, I answer some of the the authors’ responses that need further clarification. The numbers refer to the original comment numbering.
Comment 2:
This still does not explain why the use of hyperspectral imagery is needed in this application. It is obvious that hyperspectral data may allow distinguishing objects that cannot be distinguished in multispectral or RGB imagery. But as mentioned in my previous comment, you have not clarified why using hyperspectral imagery in this specific application is even needed. The example images you provide look as if they could be reliably analyzed based on multispectral data (VIS, possibly NIR). Please provide a reasoning for the use of hyperspectral imagery.
Comment 3d:
It is important to know how many clusters DBSCAN distinguishes to give the reader a direct comparison with the other algorithms where the number of clusters is fixed. As mentioned before, the adaptive, unsupervised selection of the number of clusters might be one reason (among many) why DBSCAN outperforms the other two methods. Please provide the cluster count of your DBSCAN experiments and how it changes with changing EPS.
Comment 12:
I think you deleted but not replaced some words. Please check the sentences for correctness.
